# Shikonin Derivatives from *Onsoma visianii* Decrease Expression of Phosphorylated STAT3 in Leukemia Cells and Exert Antitumor Activity

**DOI:** 10.3390/nu13041147

**Published:** 2021-03-31

**Authors:** Zeljko Todorovic, Jelena Milovanovic, Dragana Arsenijevic, Nenad Vukovic, Milena Vukic, Aleksandar Arsenijevic, Predrag Djurdjevic, Marija Milovanovic, Nebojsa Arsenijevic

**Affiliations:** 1Department of Internal Medicine, Faculty of Medical Sciences, University of Kragujevac, 34000 Kragujevac, Serbia; todorovic_zeljko@hotmail.com (Z.T.); pdjurdjevic@sbb.rs (P.D.); 2Center for Molecular Medicine and Stem Cell Research, Faculty of Medical Sciences, University of Kragujevac, 34000 Kragujevac, Serbia; jelenamilovanovic205@gmail.com (J.M.); menki@hotmail.rs (D.A.); aleksandar@medf.kg.ac.rs (A.A.); arne@medf.kg.ac.rs (N.A.); 3Department of Histology and Embriology, Faculty of Medical Sciences, University of Kragujevac, 34000 Kragujevac, Serbia; 4Department of Pharmacy, Faculty of Medical Sciences, University of Kragujevac, 34000 Kragujevac, Serbia; 5Department of Chemistry, Faculty of Science, University of Kragujevac, 34000 Kragujevac, Serbia; nvukovic@kg.ac.rs (N.V.); milena.vukic@kg.ac.rs (M.V.)

**Keywords:** isobutyrylshikonin, α-methylbutyrylshikonin, *Onosma visianii*, STAT3, BCL1, JVM-13

## Abstract

Antitumor effects of shikonins on chronic lymphocytic leukemia (CLL) and B-cell prolymphocytic leukemia (B-PLL) are mostly unexplored. The antitumor activity of shikonins, isolated from *Onosma visianii* Clem (*Boraginaceae*), in BCL1, mouse CLL cells and JVM-13, human B-PLL cells was explored in this study. The cytotoxicity of shikonin derivatives was measured by an MTT test. Cell death, proliferation, cell cycle, and expression of molecules that control these processes were analyzed by flow cytometry. Expression of STAT3-regulated genes was analyzed by real-time q-RT-PCR (Quantitative Real-Time Polymerase Chain Reaction). The antitumor effects of shikonin derivatives in vivo were analyzed, using flow cytometry, by detection of leukemia cells in the peripheral blood and spleens of mice intravenously injected with BCL1 cells. The two most potent derivatives, isobutyrylshikonin (IBS) and α-methylbutyrylshikonin (MBS), induced cell cycle disturbances and apoptosis, inhibited proliferation, and decreased expression of phospho-STAT3 and downstream-regulated molecules in BCL1 and JVM-13 cells. IBS and MBS decreased the percentage of leukemia cells in vivo. The link between the decrease in phosphorylated STAT3 by MBS and IBS and BCL1 cell death was confirmed by detection of enhanced cell death after addition of AG490, an inhibitor of Jak2 kinase. It seems that IBS and MBS, by decreasing STAT3 phosphorylation, trigger apoptosis, inhibit cell proliferation, and attenuate leukemia cell stemness.

## 1. Introduction

Shikonin, a natural naphthoquinone pigment, first isolated from *Lithospermum erythrorhizon*, a Chinese herbal plant, was used for thousands of years in traditional Chinese medicine [1]. Recently, shikonin and its derivatives, have been isolated from the root of *Onosma visianii* Clem (*Boraginaceae*), a biennial to perennial plant and an inhabitant of the Balkan Peninsula and Southeast Europe [2]. Anti-inflammatory, antimicrobial, antioxidant, antitumor, and tissue reparative activities of shikonins have been reported [2,3]. Biological activities of shikonins are mediated by several molecules, including signal transducer and activator of transcription 3 (STAT3) [3]. STAT3, by relaying signals from activated growth and cytokine receptors in the plasma membrane to the nucleus, regulates the expression of numerous genes that control cell proliferation, differentiation, apoptosis, self-renewal, angiogenesis, and immune response, which collectively contribute to malignant transformations [4]. Constitutive activity of STAT3 has been reported in a variety of hematopoietic and solid malignancies including chronic lymphocytic leukemia (CLL) [5]. It has been reported that STAT3 inhibition induces tumor cell death and increases apoptosis of tumor cells by increasing the Bax/Bcl-2 ratio [6]. High expressions of stem-cell-specific molecules (Oct4, Nanog, and Sox2) regulated by STAT3 have been reported in several malignancies and are accompanied with the stem phenotype of cancer cells [7].

The incidence of CLL in Eastern Europe and North America is approximately 4.7 per 100,000 people per year, while, in the population over 85 years of age, the CLL incidence is higher than 35 [8]. The use of novel targeted therapies, like Bruton tyrosine kinase inhibitors and inhibitors of Bcl2 family proteins, have improved the prognosis of CLL patients, including elderly patients and those with poor risk features [9]. However, CLL is still an incurable and progressive disease. Stimulation of the B-cell receptor (BCR) in CLL cells leads to phosphorylation of STAT3, indicating STAT3 as a novel target for therapeutic intervention in CLL [10]. B-cell prolymphocytic leukemia (B-PLL) is, like CLL, a chronic lymphoproliferative disorder but with a much lower incidence, accounting for less than 1% of mature B cell malignancies [11]. Initially described as a variant of CLL [12], B-PLL was firstly recognized as a distinct mature B cell entity in the World Health Organization classification in 2008 [11]. B-cell prolymphocytic leukemia is a malignancy that is common in the elderly [13] and is an incurable disease with an even lower survival rate than CLL.

Antibacterial as well as cytotoxic activities against human colorectal and breast cancer cell lines of seven shikonin derivatives isolated from *Onosma visianii* (deoxyshikonin, isobutyrylshikonin (IBS), α-methylbutyrylshikonin (MBS), acetylshikonin, β-hydroxyisovalerylshikonin, 5,8-O-dimethyl IBS and 5,8-O-dimethyl deoxyshikonin) have recently been reported [2]. However, the mechanism of cytotoxic activity has not been reported. The aim of this study was to explore the antitumor activity of these shikonin derivatives in murine BCL1 chronic lymphocytic leukemia cells and in human JVM-13 B-cell prolymphocytic leukemia to delineate a possible mechanism of action.

## 2. Materials and Methods

### 2.1. Chemicals

Shikonin derivatives were isolated and purified from *Onosma visianii* as previously described [2]. For in vitro experiments, all derivatives were first dissolved in dimethyl sulfoxide (DMSO) (Sigma-Aldrich) in such a way that the final concentration of DMSO in the medium was never greater than 0.5%. For in vivo research, derivatives were dissolved in olive oil. Cyclophosphamide (Sigma-Aldrich, Saint Louis, MO,USA) was used as control substances for in vivo experiments.

### 2.2. Cell Line

A BCL1 mouse CLL cell line and a JVM-13 human B-PLL cell line were obtained from the American Type Culture Collection (ATCC-LGC, Wesel, Germany). BCL1 cells were cultured in RPMI 1640 supplemented with 2 mM glutamine, 0.05 mM 2-2-mercaptoethanol, 100 units/mL penicillin, 100 μg/mL streptomycin, and 15% fetal bovin serum (FBS). JVM-13 cells were cultured in RPMI 1640 with 2 mM glutamine, 100 units/mL penicillin, 100 μg/mL streptomycin, and 10% FBS. All cells were grown in a 5% CO_2_ incubator with standard conditions.

### 2.3. MTT Assay

A 6 × 10^3^ cells/well was seeded in a 96-well plate, and 100 μL solutions of different naphthoquinone compounds were added in final concentrations from 0.5 μg/mL to 100 μg/mL. At the end of 24 or 48 h treatment, MTT (3-(4,5-dimethylthiazol-2-yl)-2,5-diphenyltetrazolium bromide) solution (5 mg/mL) was added to each well and incubated for an additional 4 h. The absorbance was measured at 595 nm using a Zenyth 3100 microplate multimode detector. Cell viability was measured as the percent ratio of absorbance in shikonin-derivative-treated cells over the control.

### 2.4. Assessment of Cell Death by Flow Cytometry

After treatment with IBS and MBS for 24 h at concentrations of 1 μg/mL and 2 μg/mL, BCL1 and JVM-13 cells were stained with Annexin V-FITC and propidium iodide (PI) (BD Pharmingen, San Diego, CA, USA) according to the manufacturer’s instructions. The percentages of late and early apoptotic cells were determined using a FACSCalibur flow cytometer (BD Biosciences, San Jose, CA, USA), and the data were analyzed using FlowJo (Tree Star).

### 2.5. Cell Cycle Analysis

After 24 h treatment with IBS and MBS at concentrations of 1 μg/mL and 2 μg/mL, BCL1 and JVM-13 cells were stained with Vybrant^®^ DyeCycle™ Ruby stain according to the manufacturer’s instructions. Cell cycle distribution was analyzed by the FACSCalibur flow cytometer (BD Biosciences, San Jose, CA, USA), and the data were processed using FlowJo (Tree Star).

### 2.6. Flow Cytometric Analysis

BCL1 and JVM-13 cells, after IBS and MBS treatment for 24 h, were fixed, permeabilized, and incubated with antibodies specific for Tyr 705 phosphorylated STAT3 (sc-8059, 200 µg/mL, Santa Cruz Biotech. Inc., Santa Crus CA, USA, dilution 1:300) and Ki-67 (11-5698-82, 100 µg/mL, eBioscience, San Diego, CA, USA, dilution 1:400). JVM-13 cell were stained with anti-cyclin D3 (ab28283, 100 µg/mL, Abcam Cambridge, UK, dilution 1:100) antibody. BCL1 cells were stained with anti-p21 (ab188224, 100 µL, Abcam Cambridge, UK, dilution 1:50), anti-p16 (ab211542, 100 µL, Abcam Cambridge, UK, dilution 1:500), and anti-27 antibodies (ab215434, 100 µL, Abcam Cambridge, UK, dilution 1:100). Cells were additionally incubated with secondary goat anti-mouse IgG FITC (ab6717-1, 1 mg/mL, Abcam Cambridge, UK, dilution 1:2000). The FACSCalibur flow cytometer (BD Biosciences, San Jose, CA, USA) was used for flow cytometric analysis, and the data were analyzed using FlowJo (Tree Star).

### 2.7. Immunofluorescence Staining

The expression of Bax and cleaved caspase-3 proteins was investigated by the immunofluorescence method. The BCL1 cells were seeded in a 6-well plate and exposed to MBS at a concentration of 2 mg/mL for 24 h. After washing the cells twice with PBS (Phosphate Buffered Saline), they were fixed in 4% paraformaldehyde at 25 °C for 20 min. The cells were stained with a rabbit polyclonal antibody specific for Bax (sc-493, 100 µg/mL, Santa Cruz Biotech. Inc., CA, USA, dilution 1:1000) and active/cleaved caspase-3 (NB100-56113, 250 µL, Novus Biologicals, Abingdon, UK, dilution 1:1000). After incubation, the cells were washed and treated with an appropriate secondary antibody, goat anti-rabbit IgG FITC (ab6717-1, 1 mg/mL, Abcam, Cambridge, UK, dilution 1:2000). The sections were mounted with ProLong Gold antifade reagent with (4′,6-diamidino-2-phenylindole) DAPI (Invitrogen) and analyzed at × 200 magnification using a fluorescent microscope (Olympus BX 51).

### 2.8. RNA Extraction and Real-Time qRT-PCR

After 24 h treatment with IBS and MBS at a concentration of 2 μg/mL, RNA was isolated from BCL1 and JVM-13 cells with TRIzol (Invitrogen, Carlsbad, CA, USA). Reverse transcription was performed on isolated RNA with a High-Capacity cDNA Reverse Transcription Kit (Applied Biosystems, Foster City, CA, USA). qRT-PCR was performed using Power SYBR MasterMix (Applied Biosystems) and mRNA-specific primers (Table 1). qPCR reactions were initiated with a 10 min incubation time at 95 °C followed by 40 cycles of 95 °C for 15 s and 60 °C for 60 s in a Mastercycler ep realplex (Eppendorf, Hamburg, Germany). Relative expression of genes was calculated according to the formula 2^−(Ct−Ctactin)^, where C_t_ is the cycle threshold of the gene of interest and C_tactin_ is the cycle threshold value of the housekeeping gene (β-actin).

### 2.9. In Vivo Testing of Antitumor Activity

Male BALB/c mice, aged 8–10 weeks old, were used for the experiments. All animals received humane care, and all experiments were approved by and conducted in accordance with the guidelines of the Animal Ethics Committee of the Faculty of Medicine, Kragujevac, Serbia. Mouse models of leukemia were conducted by intravenous injection of 1 × 10^6^ BCL1 cells into the tail vein. Twenty days after BCL1-cell inoculation, peripheral blood was tested by flow cytometry and mice with an increase in CD5^+^CD19^+^ cells were randomly divided into six groups: (1) IBS, 2 mg/kg; (2) IBS, 4 mg/kg; (3) MBS 2 mg/kg; (4) MBS 4 mg/kg; (5) cyclophosphamide once weekly for 2 consecutive weeks at a dose of 100 mg/kg (maximal tolerable dose for BALB/c mice [14]); and (6) saline. The treatment with shikonins (intraperitoneal) lasted for two weeks and each mouse received six doses in total.

All mice were sacrificed two days after the last dose of shikonin derivatives. Mononuclear cells from blood and spleens incubated with anti-CD5 and anti-CD19 antibodies were analyzed by flow cytometry, and the data were analyzed using FlowJo (Tree Star). Serum levels of alanine amino transaminase (ALT), aspartate aminotransferase (AST), urea, and creatinine were measured by a standard photometric method using the Olympus AU 400 automated biochemistry analyzer (Olympus Diagnostica GMBH, Hamburg, Germany) and Olympus AU reagents, according to the manufacturer’s instructions.

### 2.10. Statistical Analysis

The data are presented as means ± SDs. Statistical significance was determined by an independent sample Student *t*-test and ANOVA, and, where appropriate, a Mann–Whitney *U*-test or a Kruskal–Wallis test. Statistical significance was assumed at *p* < 0.05. Statistical analyses were performed using SPSS 13.0.

## 3. Results

### 3.1. Shikonin Derivates Significantly Reduce the Viability of BCL1 and JVM-13 Cells

BCL1 and JVM-13 cell viability after treatment with growing concentrations of five different derivatives of shikonin, IBS, MBS, acetylshikonin, β-hydroxyisovalerylshikonin, and deoxyshikonin, for 24 and 48 h was analyzed by MTT assay and IC50 values were calculated. The obtained data did not show reduced viability of BCL1 and JVM-13 cells treated with β-hydroxyisovalerylshikonin and deoxyshikonin (Table 2). IC50 values for acetylshikonin after 24 h and 48 h of exposure indicated good cytotoxicity toward BCL1 and JVM-13 cells (Table 2). However, IBS and MBS reduced the viability of BCL1 and JVM-13 cells up to 50% in extremely low concentrations (Table 1). The cytotoxic effect of these derivatives was not time-dependent, and IC50 values after 24 and 48 h of exposure were nearly the same.

As shown in Figure 1, IBS and MBS decreased the viability of BCL1 and JVM-13 cells in a dose-dependent manner. From a concentration of 3 μg/mL, both IBS and MBS reduced the viability of BCL1 cells to 100% (Figure 1a). A similar effect was obtained on the JVM-13 cell line at a concentration of 10 μg/mL and higher for both shikonin derivatives (Figure 1b).

### 3.2. IBS and MBS Induce Apoptosis of BCL1 and JVM-13 Cells

In order to further investigate the antitumor potential of shikonin derivatives with the strongest potential to reduce the viability of BCL1 and JVM-13 cells, flow cytometric analysis of cells stained with Annexin V and PI after exposure to IBS and MBS for 24 h was done. As illustrated in Figure 2, IBS- and MBS-induced apoptosis and the majority of BCL1 and JVM-13 cells (Figure 2a) were in or occurred during late apoptosis. The percentage of late apoptotic cells was significantly higher (*p* < 0.005) after treatment with IBS and MBS at the 2µg/mL concentration compared to untreated cells in both BCL1 (Figure 2a,b) and JVM-13 (Figure 2a) cell lines. Additionally, the percentage of late-apoptotic cells of both cell lines treated with lower concentrations of IBS and MBS was significantly lower in comparison with the percentage of cells treated with higher concentrations of tested derivatives of shikonin, showing that the pro-apoptotic effect of the substances is dose-dependent (Figure 2a). Treatment with IBS and MBS increased the percentage of early apoptotic cells as well, but percentages were lower compared to late apoptotic cells in both BCL1 and JVM-13 (Figure 2a) cell lines. Quantitative analysis of Bcl-2 and Bax and Mcl-1 and Noxa molecules in JVM-13 cells (Figure 2b) strongly indicated apoptotic death of JVM-13 cells after treatment with MBS and IBS. Treatment of JVM-13 cells with IBS and MBS at a concentration of 2 µg/mL significantly attenuated the percentage of Mcl-1-expressing cells and significantly enhanced the percentage of cells that expressed the pro-apoptotic molecule Noxa, even at a lower concentration of 1 µg/mL (Figure 2c). Analysis of Bax and Bcl-2 expression at mRNA levels also showed significant attenuation of Bcl*-2* and enhancement of Bax mRNA levels after treatment of JVM-13 cells with IBS and MBS (Figure 2d). Similarly, tested shikonins at a concentration of 2 µg/mL significantly attenuated the percentage of Mcl-1 expressing BCL1 cells and enhanced the percentage of Noxa positive BCL1 cells (Figure 2e). mRNA levels of Bax were significantly higher in BCL1 cells treated with MBS and IBS (Figure 2c), while the mRNA levels of cleaved caspase-3 were also higher in treated cells but did not reach statistical significance (Figure 2f). In accordance with this finding, the expression of both pro-apoptotic molecules, Bax and cleaved caspase-3, at protein levels increased after MBS treatment (compared to untreated BCL1 cells) as evaluated by immunofluorescence staining (Figure 2g).

### 3.3. Derivatives of Shikonin Show Antiproliferative Effect against BCL1 Cells

The anti-proliferative effects of testing molecules were evaluated by assessment of the Ki67 expression level in treated cells using flow cytometry. The percentage of Ki67 positive BCL1 and JVM-13 cells (Figure 3a) treated with IBS and MBS was significantly lower compared to untreated cells. Additionally, lower expressions of Ki67 were found in treated IBS and MBS compared to untreated BCL1 and JVM-13 cells (Figure 3b).

Both shikonin derivatives, in accordance with reduced Ki67 expression (Figure 3a,b), significantly reduced the percentage of BCL1 cells in the S phase of the cell cycle (Figure 4a). Treatment with IBS significantly increased the percentage of BCL1 cells in the G2/M phase compared to untreated cells (Figure 4a). Furthermore, the percentage of BCL1 cells in the G0/G1 phase decreased after IBS treatment. On the other hand, MBS treatment significantly increased the percentage of BCL1 cells in the G0/G1 phase compared to untreated cells and decreased the percentage of these cells in the G2/M phase (Figure 4a). However, both tested derivatives of shikonin significantly increased the percentage of JVM-13 cells in the G0/G1 phase and decreased the percentage of these cells in the S and G2/M phases (Figure 4b). A significant reduction in the expression of cyclin D3 mRNA was detected in IBS- and MBS-treated BCL1 cells in comparison with untreated cells (Figure 4c). Additionally, a significant decrease in the percentage of cyclin-D3-expressing JVM-13 cells after treatment with IBS (2 µg/mL), and with both tested concentrations of MBS, was noticed (Figure 4d). The percentage of BCL1 cells expressing the inhibitor of cyclin D-CDK4 complex, p16, was significantly enhanced after treatment with IBS and MBS compared to untreated BCL1 cells (Figure 4e). No significant change in the expression of p21 and p27 BCL1 cells treated with IBS was noticed (Figure 4e). However, the treatment of BCL1 cells with MBS significantly increased the percentage of p21- and p27-expressing cells compared to untreated BCL1 cells (Figure 4e).

### 3.4. Shikonin Derivatives Inhibit Expression of pSTAT3 and STAT3 Regulated Genes

Since it has been shown that inhibition of STAT3 signaling leads to CLL cell death [15] and that the antitumor effects of shikonins are mediated by STAT3 inhibition [16,17], the expression of pSTAT3 in IBS- and MBS-treated BCL1 and JVM-13 cells was analyzed by flow cytometry. Treatment with IBS and MBS significantly reduced the percentage of pSTAT3-expressing BCL1 and JVM-13 cells compared to untreated cells (Figure 5a). This was accompanied by a significantly lower expression of c-Myc, Nanog, and Oct4 mRNA in BCL1 cells and a lower expression of c-Myc, Nanog, Sox2, and Oct4 mRNA in JVM-13 cells (Figure 5b). To further explore the role of shikonin-induced STAT3 inhibition in tumor cell death, the cytotoxic activity of combination of IBS and MBS with the single agent inhibitor of Jak2 (AG490), a molecule that phosphorylates STAT3, was analyzed by MTT assay. As shown in Figure 5c, AG490 alone does not reduce the viability of BCL1 cells after 24 h of exposure. However, simultaneous treatment with AG490 and IBS or MBS induces a significantly higher reduction in BCL1 cell viability in comparison with the single treatment (Figure 5c).

### 3.5. Shikonin Derivatives Reduced BCL1 Cell Growth In Vivo

According to results obtained by the MTT test, we investigated the potential of IBS and MBS to reduce the growth of leukemia cells in BALB/c mice intravenously injected with BCL1 cells. Cyclophosphamide was used as a control substance. Treatment with two different doses of IBS and MBS (2 mg/kg and 4 mg/kg) significantly reduced the percentage of CD5+CD19+ cells in the peripheral blood and spleen of BCL1-injected BALB/c mice (Figure 6a,b). A significantly higher dose of IBS and MBS (4 mg/kg) led to a decrease in the percentage of leukemic cells in the spleen similar to the decrease induced by cyclophosphamide treatment (Figure 6b). Importantly, IBS and MBS treatment did not significantly change the serum level of ALT, AST, urea, and creatinine (Figure 6c).

## 4. Discussion

CLL is a slowly progressive but still incurable disease due to the presence of apoptosis-resistant lymphocytes that proliferate in bone marrow and lymph nodes [18]. Therefore, exploring new treatment options capable to induce apoptosis in CLL cells is necessary. In this study, we provide the first evidence that isobutyrylshikonin and α-methylbutyrylshikonin, two naphthoquinones extracted from the roots of *Onosma visianii*, induce strong apoptotic, antiproliferative, and STAT3-dependent cytotoxic effects in mouse CLL cells, BCL1, and human B-PLL cells, JVM-13, with significant in vivo antitumor effects.

Shikonin is a well-known antitumor substance that induces apoptosis, inhibits proliferation of cancer cells, and inhibits angiogenesis. The antitumor effects of shikonin have been described in various tumors including leukemia [19]. Previous reports indicate that root extracts and several shikonin derivatives isolated from the endemic plant of the Balkan peninsula, *Onosma visianii,* exert antitumor effects in vitro [2,20] but also bind human albumin well [21], suggesting that they could be effectively transported by serum albumin and could have antitumor effects in vivo. In this study, we confirmed that two shikonin derivatives, IBS and MBS, significantly reduced viability, suppressed proliferation, and induced apoptotic death in BCL1 and JVM-13 cells in a dose-dependent manner (Figure 1, Figure 2 and Figure 3). This finding is in line with findings from previous testing of naphthoquinones isolated from *Onosma visianii*, which revealed that IBS significantly reduced the viability of human colorectal cancer cells [2]. Here, we have also shown that the therapeutic application of IBS and MBS in mice with BCL1 leukemia induced marked antitumor action as evaluated by a reduced number of BCL1 cells counted in peripheral blood (Figure 6a) and spleen (Figure 6b). Additionally, an important observation is that both IBS and MBS induced a decrease in tumor cells in the spleens almost as effective as cyclophosphamide (Figure 6b).

In line with previous findings that shikonin induce apoptosis in tumor cells [22,23], we found that IBS and MBS induce apoptotic death in BCL1 and JVM-13 treated cells (Figure 2). The majority of BCL1 and JVM-13 cells treated with both doses of IBS and MBS were late apoptotic (Figure 2a), suggesting very prompt pro-apoptotic effects. Furthermore, the high expression of molecules that play an important role in the apoptotic cell death, Bax and cleaved caspase-3 [24], was observed in MBS-treated BCL1 cells (as evaluated by immunofluorescence) (Figure 2g). In line with these findings, treatment with IBS and MBS upregulate mRNA levels of Bax and caspase-3 (Figure 2f). A decrease of anti-apoptotic Mcl-1 and an increase of pro-apoptotic Noxa was also observed in both cell lines (Figure 2c,e), proteins whose relative ratio controls apoptosis [25]. Our results suggest that IBS and MBS induce apoptosis in BCL1 and JVM-13 cells and are in accordance to previous findings that shikonin induces a decrease in Bcl-2 and an increase in Bax expression, enhances the activities of caspase-3, and induces apoptotic cell death [26,27].

A recent study showed that shikonin could suppress cell proliferation via arresting the cell cycle at the G0/G1 or the G2/M phase [2]. We have shown that both derivatives increased the percentage of JVM-13 cells in the G0/G1 phase (Figure 4b), while the percentage of BCL1 cells in the G0/G1 phase was increased after MBS treatment (Figure 4a). Cyclin D3 binds to CDK4 and CDK6 and plays a role as the initial activator of the G1 phase [28,29,30,31]. In line with this, MBS treatment induced a decrease in the mRNA level of cyclin D3 in BCL1 cells (Figure 4c) and the percentage of cyclin D3 expressing JVM-13 (Figure 4d) but significantly enhanced the expression of the protein level of the inhibitor of cyclin D–CDK4/CDK6 complex, p16 [28], the universal cyclin-CDK inhibitor, p21 [29], and the inhibitor of cyclin E-CDK2, cyclin A-CDK2, and cyclin D-CDK4 complexes, p27 [30], in BCL1 cells (Figure 4e). IBS and MBS significantly decreased the percentage of BCL1 and JVM-13 cells expressing Ki67 (Figure 3a), attenuated the expression of this molecule in both cell types (Figure 3b), and significantly decreased the percentage of these cells in the S phase of the cell cycle (Figure 4a,b). Given these results, it can be assumed that MBS inhibits BCL1 and JVM-13 cells’ proliferation by arresting them in the G0/G1 phase of the cell cycle. A similar effect of IBS was also observed in JVM-13 cells. However, IBS-treated BCL1 cells were arrested in the G2/M phase, had a lower expression of cyclin D3, and a higher expression of p16 in comparison with untreated cells (Figure 4). Previously, it has been shown that low doses of UV radiation induce G2 phase cell cycle arrest accompanied with increased expression of p16 that binds to Cdk4 and Cdk6 complexes and inhibits a cyclin D3-CDK4 complex normally activated in late S/early G2 phase [32]. Our finding of lower expression of cyclin D3, higher expression of p16, and unchanged expression of p21 and p27 in BCL1 cells (Figure 4) are in line with this finding. The arrest of BCL1 cells in different phases of the cell cycle induced by different derivatives of shikonin obtained in this study may explain the different effects of various root extracts (acetone or chloroform) on cell cycle progression previously reported, since differently obtained extracts could contain different percentages of active compounds [2].

Results from in vitro studies indicate that STAT3 inhibition increases the Bax/Bcl-2 ratio leading to apoptosis of tumor cells [6]. Additionally, STAT3 plays a key role in the G1 to S phase cell-cycle transition by the upregulation of cyclin D1, D2, D3 and the concomitant downregulation of p21 and p27 [15]. The present study revealed that MBS and IBS treatment significantly decreased the percentage of BCL1 and JVM-13 cells expressing pSTAT3 (Figure 5a). Decreased expression of pSTAT3 accompanied with increased expression of Bax, increased apoptosis of BCL1 and JVM-13 cells (Figure 2), decreased expression of cyclin D3, and increased expression of inhibitors of CDKs (Figure 4) is in agreement with previous reports [6,33].

It has recently been shown that JAK2 inhibitor, AG490, causes the dephosphorylation of STAT3 and enhances cytotoxic activity of conventional chemotherapeutics in leukemia cells [15]. To further confirm the link between decrease of pSTAT3 and enhanced death of BCL1 cells, these cells were co-treated with AG490 and MBS or IBS. As shown in Figure 5c, a significant decrease in viable BCL1 cells co-treated with AG490 and MBS or IBS in comparison with IBS or MBS treatment alone was noticed, supporting the hypothesis that IBS and MBS cytotoxic activity toward BCL1 cells is at least partially mediated by regulation of the activity of STAT3 signaling.

Aberrant expression of Nanog, Oct4, and c-Myc, downstream proteins of the STAT3 signaling pathway, was found in several cancers and is accompanied with cancer stemness [34]. In this study, we have shown that MBS and IBS treatment significantly decreased mRNA expression of Nanog, Oct4 and c-Myc in BCL1 cells and Nanog, Oct4, Sox2 and c-Myc in JVM-13 cells compared to untreated cells (Figure 5b). The correlation of the decreased percentage of leukemic cells containing Y705 phospho-STAT3 (Figure 5a) and the decreased expression of Nanog, Oct4, Sox2 and c-Myc (Figure 5b) after treatment with MBS or IBS mRNA is in line with a previous report that found inhibited phosphorylation of Stat3 decreased the expression of Oct-4 and c-Myc in breast cancer cells [16].

## 5. Conclusions

In conclusion, we reported that IBS and MBS exhibit a considerable antitumor effect in CLL and B-PLL cells. This effect was achieved by the reduction of STAT3 phosphorylation and signaling and the consecutive triggering of apoptotic cell death, inhibition of cell proliferation, and attenuation of cancer stemness. These results highlight the necessity of further testing of shikonin derivatives as possible new anticancer agents or auxiliary drugs.

## Figures and Tables

**Figure 1 nutrients-13-01147-f001:**
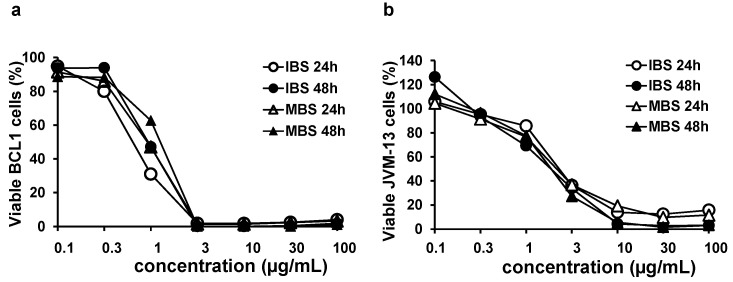
Dose- and time-dependent cytotoxicity of isobutyrylshikonin (IBS) and α-methylbutyrylshikonin (MBS) on the BCL1 cell line. Graphs of (**a**) BCL1 and (**b**) JVM-13 cells’ survival after 24 and 48 h growth in the presence of IBS and MBS determined by MTT assay. All data are presented as mean values from three independent experiments performed in triplicate.

**Figure 2 nutrients-13-01147-f002:**
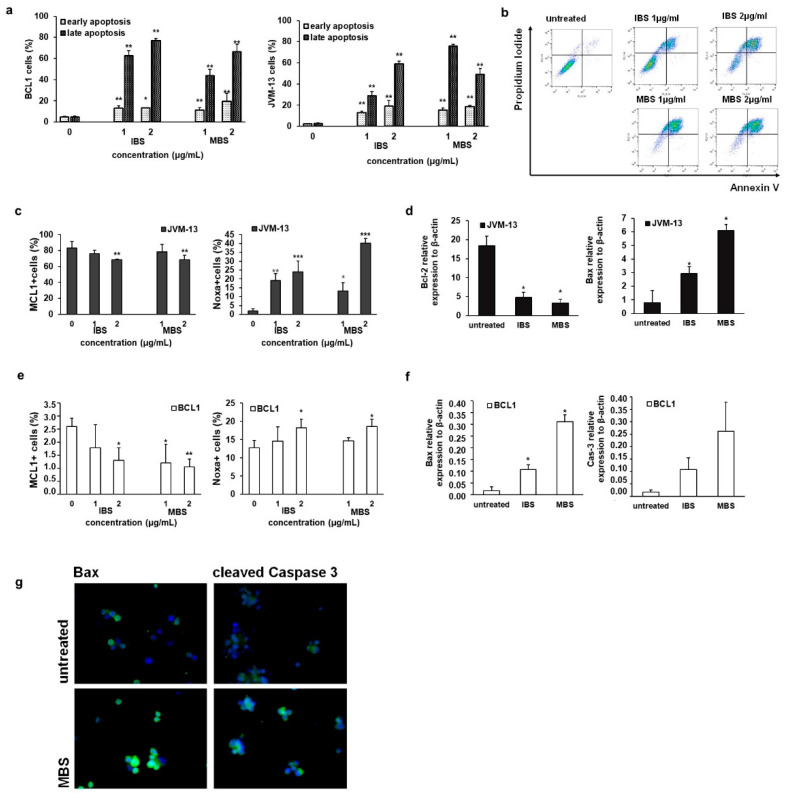
IBS and MBS induce apoptotic death of leukemia cells. Apoptosis of untreated and IBS- and MBS-treated BCL1 and JVM-13 cells for 24 h (**a**) evaluated by flow cytometry using Annexin V (FITC) and PI double staining. (**b**) Representative dot plots illustrate populations of viable (AnnV− PI-), early apoptotic (Ann V+ PI−), late apoptotic (AnnV+ PI+), and necrotic (AnnV− PI+) BCL1 cells treated with IBS or MBS. Percentage of Mcl-1 and Noxa positive (**c**) JVM-13 and (**e**) BCL1 cells evaluated by flow cytometry. mRNA expression of Bax and Bcl*-2* in (**d**) JVM-13 and Bax and (**f**) caspase-3 expression in BCL1 cells treated by IBS and MBS quantified by qRT-PCR. (**g**) Immunofluorescence staining for Bax (green) and cleaved caspase-3 (green) together with DNA staining with DAPI (blue) in BCL1 cells incubated with MBS (2 µg/mL) for 24 h, as well as in untreated cells (magnification × 200). The data are presented as means + SDs of a three independent experiment. * *p* < 0.05, ** *p* < 0.01, *** *p* < 0.001 indicate differences between treated and untreated cells.

**Figure 3 nutrients-13-01147-f003:**
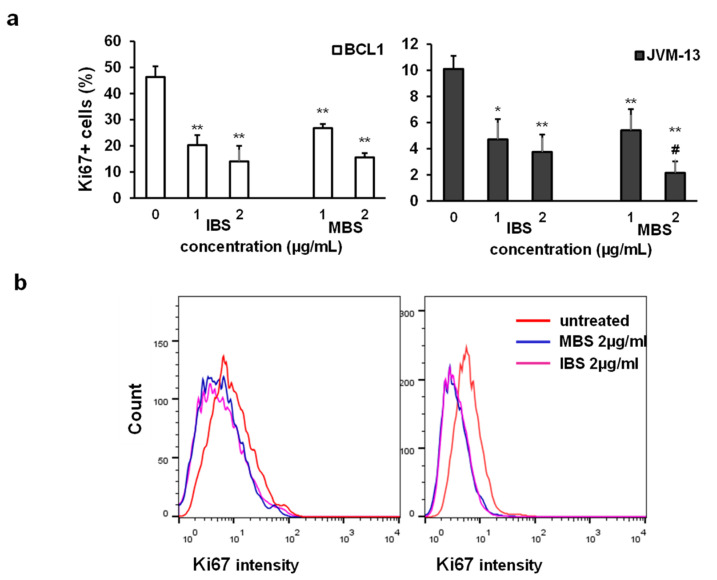
IBS and MBS attenuate the expression of Ki67 in BCL1 and JVM-13 cells. (**a**) Percentage of Ki-67 positive BCL1 and JVM-13 cells exposed to IBS and MBS (concentrations = 1 and 2 µg/mL) for 24 h determined by flow cytometry presented as the mean + SD from three independent experiments. Data were analyzed with Student’s *t*-test: * *p* < 0.05; ** *p* < 0.01, # *p* < 0.05 (indicates differences between two doses of MBS). (**b**) Representative histograms of Ki67 expression (mean fluorescence intensity) in BCL1 and JVM-13 cells.

**Figure 4 nutrients-13-01147-f004:**
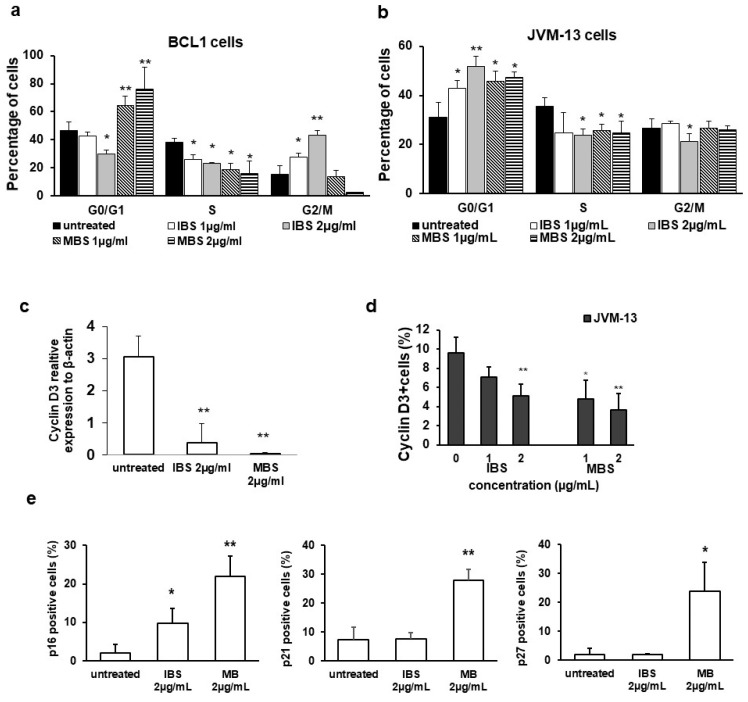
IBS and MBS induce cell cycle disturbances in JVM-13 and BCL1 cells. Distribution of (**a**) BCL1 and (**b**) JVM-13 cells in different phases of the cell cycle after treatment with IBS and MBS (concentrations = 1 µg/mL and 2 µg/mL) was determined by flow cytometry after staining with Vybrant^®^ DyeCycle™ Ruby stain. Results are expressed as the percentage of cells in different phases of the cell cycle. (**c**) mRNA expression of cyclin D3 quantified by RT-PCR in BCL1 cells after treatment with IBS and MBS (2 µg/mL) for 24 h. (**d**) Percentage of JVM-13 cells expressing cyclin D3 after treatment with two different concentrations of IBS and MBS (1 µg/mL and 2 µg/mL) for 24 h determined by flow cytometry. (**e**) Percentage of p16-, p21-, and p27-positive BCL1 cells analyzed by flow cytometry after 24 h treatment with IBS and MBS. Data are presented as the mean ± SD. * *p* < 0.05, ** *p* < 0.01 indicates a significant difference from untreated cells.

**Figure 5 nutrients-13-01147-f005:**
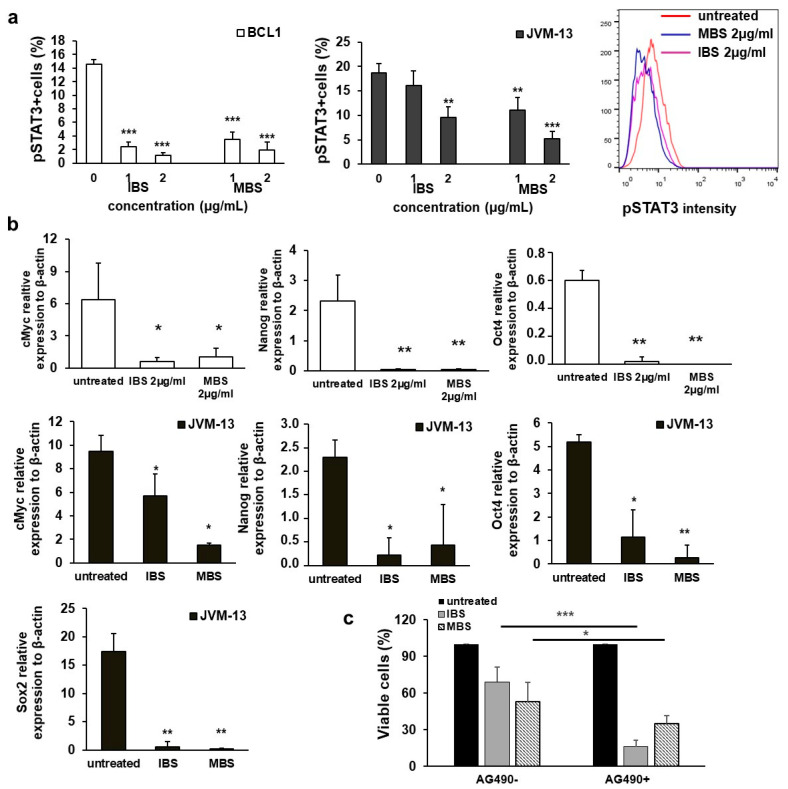
Treatment with IBS and MBS attenuates the expression of pSTAT3- and STAT3-regulated genes in BCL1 and JVM-13 cells. (**a**) Percentages of pSTAT3 positive BCL1 and JVM-13 cells determined by flow cytometry 24 h after treatment with IBS and MBS and representative histograms of pSTAT3 expression in JVM-13 cells. (**b**) mRNA expression of c-Myc, Nanog, and Oct4 in BCL1 cells and c-Myc, Nanog, Sox2 and Oct4 in JVM-13 cells treated with IBS and MBS quantified by RT-PCR. (**c**) BCL1 cells’ survival after 24 h growth in the presence of IBS and MBS, with or without the addition of Jak2 inhibitor AG490 (tyrphostin), determined by MTT assay. Data are presented as the mean ± SD. * *p* < 0.05, ** *p* < 0.01, *** *p* < 0.001.

**Figure 6 nutrients-13-01147-f006:**
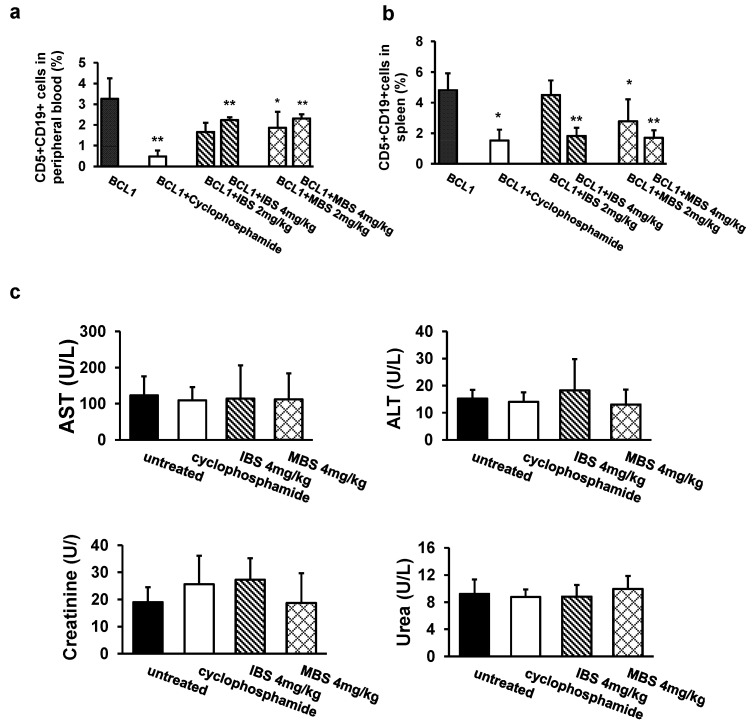
Treatment with IBS and MBS inhibits BCL1 leukemia growth in vivo. Mononuclear cells from (**a**) peripheral blood and (**b**) spleens of untreated and IBS- and MBS-treated mice 20 days after intravenous BCL1 cell injection (1 × 10^6^). The percentage of CD5+CD19+ BCL1 cells is presented as the mean ± SD. * *p* < 0.05, ** *p* < 0.01. (**c**) AST, ALT, urea, and creatinin levels were determined in the serum 90 days after the BCL1 injection in untreated and IBS- and MBS-treated mice, presented as the mean ± SD (eight animals per group).

**Table 1 nutrients-13-01147-t001:** Primers used for real-time qRT-PCR.

Target	Sense and Antisense
Mouse Bax	5’-ACACCTGAGCTGACCTTG-3´ 5´-AGCCCATGATGGTTCTGATC-3´
Mouse caspase-3	5´-AAATTCAAGGGACGGGTCAT-3´ 5´-ATTGACACAATACACGGGATCTGT-3´
Mouse cyclin D3	5´- CCGTGATTGCGCACGACTTC-3´ 5´-TCTGTGGGAGTGCTGGTCTG-3´
Mouse Nanog	5´-AAGCAGAAGATGCGGACTGT-3´ 5´-GTGCTGAGCCCTTCTGAATC-3´
Mouse Oct4	5´-CAAGGCAAGGGAGGTAGACA-3´ 5´-ATGAGTGACAGACAGGCCAG-3´
Mouse c-Myc	5´-CGGACACACAACGTCTTGGAA-3´ 5´-AGGATGTAGGCGGTGGCTTTT-3´
Mouse β-actin	5´-AGCTGCGTTTTACACCCTTT-3´ 5´-AAGCCATGCCAATGTTGTCT -3´
Human Bax	5´-ATGGACGGGTCCGGGGAGCA-3´ 5´-CCCAGTTGAAGTTGCCGTCA3-3´
Human Bcl-2	5´-CTTTGAGTTCGGTGGGGTCA-3´ 5´-GGGCCGTACAGTTCCACAAA-3´
Human cMyc	5´-AAAGGCCCCCAAGGTAGTTA-3´ 5´-GCACAAGAGTTCCGTAGCTG-3´
Human Nanog	5´-ACATGCAACCTGAAGACGTGTG-3´ 5´-CATGGAAACCAGAACACGTGG-3´
Human Sox2	5´-GAGCTTTGCAGGAAGTTTGC-3´ 5´-GCAAGAAGCCTCTCCTTGAA-3´
Human Oct4	5´-ACATCAAAGCTCTGCAGAAAGAACT-3´ 5´-CTGAATACCTTCCCAAATAGAACCC-3´
Human β-actin	5´-CACCATTGGCAATGAGCGGTTC-3´ 5´-AGGTCTTTGCGGATGTCCACGT-3´

**Table 2 nutrients-13-01147-t002:** The IC50 values of shikonins determined by MTT assay on BCL1 cell line.

Compound	IC_50_ ± SD (μg/mL)	
	24 h	48 h
Isobutyrylshikonin (IBS)	0.86 ± 0.16	0.95 ± 0.15
α-methylbutyrylshikonin (MBS)	1.07 ± 0.19	1.02 ± 0.20
Acetylshikonin	3.82 ± 0.20	3.88 ± 0.20
β-hydroxyisovalerylshikonin	≥100	≥100
Deoxyshikonin	≥100	≥100

## Data Availability

Data described in the manuscript will be made available upon reasonable request from the corresponding author.

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
