# Peer review of "Shikonin Derivatives from Onsoma visianii Decrease Expression of Phosphorylated STAT3 in Leukemia Cells and Exert Antitumor Activity"

_nutrients, 2021, doi:10.3390/nu13041147_

Round 1

Reviewer 1 Report

In this work, Zeljko Todorovic at al. evaluate the anti-leukemic potential of the plant-derived shikonins isobutyrylshikonin (IBS) and α-methylbutyrylshikonin (MBS). Overall, their results suggest that both, IBS and MBS, exert anti-tumor activity by promoting apoptosis, cell cycle disruption and decrease of pSTAT3 in leukemia/lymphoma in vitro and  in vivo models. Nonetheless, there are still some unclear important points that authors need to clarify.

Specific remarks:

Figure 2: The authors determine that BCL2 mRNA expression is reduced upon treatment in JVM-13 cell line, but no information regarding BCL1 cell line is shown. Moreover, BCL2 protein levels should be evaluated in both lines by western blot or intracellular flow cytometry analysis. Was caspase-3 evaluated in JVM-13 cell line?

Figure 4: Figure 4B missing legend. According to dose-response curve in Figure 1, 2ug/ml of IBS and MBS highly decreased BCL1 and JVM-13 viability. In this context, it could be hard to evaluate whether the compounds directly affect cell cycle or if it is just a consequence of the low viability. A dose-response curve and cell cycle analysis should be performed to clarify this point.

Figure 6: How does IBS and MBS treatment affect healthy counterparts in peripheral blood? Does the compounds reduce the viability of T and NK cells?. This point should be clarified.

Neither BCL1 (leukemia/lymphoma cell line) or JVM-13 (a B-prolymphocytic leukemia cell line) are considered exactly CLL-derived cell lines. The results obtained would be reinforced if CLL lines lines, such as MEC-1/2 or OSU-CLL, were employed to verify their results.

Minor remarks:

The text contains a several misspelling errors and data is poorly presented and needs to be improved

Line 55: misspelling error (phosphor-ylation)

Line 62: misspelling error (then/than)

Figure 1: Errors in X axis labels in Figure 1A and B (parentheses), and Y axis in figure 1B. Please, correct

Line 181: misspelling (apoptotic cell’s death)

Line 201: misspelling (BCL2/BCL1)

Figure 3: figure 3a includes “#” symbol in figure. Please, correct

Human genes symbols should italicised, with all letters in uppercase. Please, correct along the text.

In material and methods: please, indicate clones employed and phosphorylation site of pSTAT3

In conclusion, in my opinion this manuscript does not merit to be published in the present form in the terms of the presented set of data and thus my recommendation is major revision.

Author Response

We thank the reviewer for valuable comments.

Figure 2: The authors determine that BCL2 mRNA expression is reduced upon treatment in JVM-13 cell line, but no information regarding BCL1 cell line is shown. Moreover, BCL2 protein levels should be evaluated in both lines by western blot or intracellular flow cytometry analysis. Was caspase-3 evaluated in JVM-13 cell line?

mRNA level of BCL2 was also determined in BCL1 cells, since no significant reduction of BCL2 expression was detected after treatment with shikonins the graph was not included. BCL2 expression at the protein level was evaluated by immunofluorescence in BCL1 cells, but no significant differences between untreated and treated cells were noticed and it was not added to the figure. However, in both cell lines, we have found changes in the expression at the protein level of other pair of molecules whose balance regulates the susceptibility of the cells to apoptosis (Immunity. 2006 Jun;24(6):703-16, Neoplasia. 2007 Oct;9(10):871-81) Mcl1 and Noxa (decrease of antiapoptotic Mcl1 and increase of proapoptotic Noxa) and it is presented in Fig2c and Fig2e. Increase of Noxa positive and decrease of Mcl1 positive BCL1 and JVM-13 cells is in correlation with apoptosis detected in these cells by Annexin V staining after treatment with IBS and MBS. We did not have the correct primers for detection of caspase-3 in human cells.

Figure 4: Figure 4B missing legend. According to dose-response curve in Figure 1, 2ug/ml of IBS and MBS highly decreased BCL1 and JVM-13 viability. In this context, it could be hard to evaluate whether the compounds directly affect cell cycle or if it is just a consequence of the low viability. A dose-response curve and cell cycle analysis should be performed to clarify this point.

The legend for Fig 4a and Fig 4b is included (Distribution of (a) BCL1 and (b) JVM-13 cells in different phases of cell cycle after treatment with IBS and MBS...). We have previously done analysis of distribution of JVM-13 and BCL1 cells in different phases of cell cycle and we added the values for concentration 1µg/mL of IBS and MBS and it is presented in the new versions of Fig4a and Fig4b. The effects of IBS and MBS on both cell lines were similar for both tested concentrations.

Figure 6: How does IBS and MBS treatment affect healthy counterparts in peripheral blood? Does the compounds reduce the viability of T and NK cells?. This point should be clarified.

The aim of our study was to explore possible direct cytotoxic effects of IBS and MBS toward mouse leukemia lymphoma cells (BCL1) and human B-prolymphocytic leukemia cell line (JVM-13), we did not explore eventual modulation of immune response against leukemia cells by shikonins, and eventual contribution of antitumor immune response to complete cytotoxic effects of shikonins. Modulation of antitumor immune response by IBS and MBS in model of CLL could be explored in our next study.

Neither BCL1 (leukemia/lymphoma cell line) or JVM-13 (a B-prolymphocytic leukemia cell line) are considered exactly CLL-derived cell lines. The results obtained would be reinforced if CLL lines lines, such as MEC-1/2 or OSU-CLL, were employed to verify their results.

BCL1 cell line is a spontaneously arising murine lymphocyte leukemia and lymphoma that has served as a preclinical model for studying basic tumor biology and therapy of leukemia/lymphoma for decades (J Immunol 1979; 122: 1649–1654. J Immunol 1979; 123: 992–999. J Immunol 1983; 130: 2452–2455. Bone Marrow Transplant . 2004;33(11):1137-41.) and we also used it as a model for mouse CLL. Unfortunately we could not import MEC-1/2 cell lines (DSMZ) in Serbia. Since CLL with unfavorable chromosomal and genetic characteristics has an increased percentage of prolymphocytes (Br J Haematol . 2016 Sep;174(5):767-75.) (but insufficient for diagnosis of B-prolymphocytic leukemia) we decided to explore antitumor effects of MBS and IBS in other chronic B lymphoproliferative disorder, B-prolymphocytic leukemia cell line (JVM-13).

We clearly stated that we explored antitumor effects of shikonins on chronic lymphocytic leukemia (CLL) and B-cell prolymphocytic leukemia (B-PLL) cells (abstract and introduction).

Minor remarks:

The text contains a several misspelling errors and data is poorly presented and needs to be improved

Line 55: misspelling error (phosphor-ylation)

It was automatic breakage and it is still present in the revised version in line 55

Line 62: misspelling error (then/than)

it is corrected in revised version of the manuscript, line 62

Figure 1: Errors in X axis labels in Figure 1A and B (parentheses), and Y axis in figure 1B. Please, correct

it is corrected in revised version of the mansucript

Line 181: misspelling (apoptotic cell’s death)

it is corrected to apoptosis in revised version of the manuscript, line 209 in revised version

Line 201: misspelling (BCL2/BCL1)

it is corrected in revised version of the mansucript, line 229 in revised version

Figure 3: figure 3a includes “#” symbol in figure. Please, correct

# indicates significant difference p<0.05 between cells treated with 1µg/mL MBS and 2µg/mL MBS, and it is added in the figure legend in the revised version of the manuscript

Human genes symbols should italicised, with all letters in uppercase. Please, correct along the text.

We corrected it throughout the text and in the figure.

In material and methods: please, indicate clones employed and phosphorylation site of pSTAT3

We used antibody for Tyr 705 phosphorylated STAT3 and it is added in revised version of the mansucript

In conclusion, in my opinion this manuscript does not merit to be published in the present form in the terms of the presented set of data and thus my recommendation is major revision.

We believe that we made all suggested corrections and added explanations.

Reviewer 2 Report

I have none suggestions except of make a correction in axis x for Figure 1 a and b and add the concentrations of the antibodies In flow cytometric analysis in Material Section.

Author Response

I have none suggestions except of make a correction in axis x for Figure 1 a and b and add the concentrations of the antibodies In flow cytometric analysis in Material Section.

We corrected marks of the x axis in Fig1a and Fig1b.

We also added concentrations of the antibodies In flow cytometric analysis and Immunofluorescence staining in Material Section, lines 110-119 and 123-133.

Round 2

Reviewer 1 Report

The authors have addressed major revisions and solved related questions about their work and merits to be published in Nutrients in the present form.